# *Large-Scale Constraint Generation*
# Can LLMs Parse Hundreds of Constraints?

## Abstract

Large Language Models (LLMs) are commonly evaluated on instruction-following tasks with explicitly specified constraints, where difficulty typically arises from complex reasoning or long input contexts. In this work, we identify and study a distinct and largely unexplored challenge: the identification of constraint relevance. We introduce Large-Scale Constraint Generation (LSCG), a setting in which an LLM is given a large pool of valid constraints that apply in general, but must autonomously determine which subset of them is binding for a specific task instance, and then complete the task while satisfying only those constraints. Unlike existing benchmarks, the applicable constraints are not explicitly enumerated, isolating a fundamental capability required for realistic, autonomous LLM deployment.

To investigate this problem, we propose two tasks: Word Checker, which isolates constraint identification by requiring models to detect violations from long lists of forbidden words, and Language Moderator, which extends this setting to constrained generation. We evaluate multiple model families (LLaMA and R1), model sizes (8B and 70B), and inference strategies (simple prompting, Chain-of-Thought, and Best-of-N) under increasing numbers of candidate constraints.

Our results show that performance degrades sharply as constraint lists grow, dropping to near-random accuracy even for large models, indicating that scaling alone does not resolve this failure mode. Inspired by retrieval-augmented methods, we introduce FoCusNet (Focused Constraints Net), a lightweight preprocessing model that filters constraint lists to likely relevant candidates, yielding consistent accuracy improvements. Our findings highlight the identification of the relevance of the constraints as a fundamental and under-explored dimension of the following instruction.

## 1 Introduction

Instructions are natural-language prompts that guide large language models (LLMs) to perform specific tasks (e.g., "Summarize the following text") (Ouyang et al., 2022). A growing body of work has studied LLMs' ability to follow instructions that demand *complex reasoning* (Wang et al., 2023), *long-context understanding* (Bai et al., 2024; Li et al., 2024), *multi-turn coherence* (He et al., 2024c;b), or adherence to *multiple explicit constraints* (Sun et al., 2023; Yao et al., 2024; Xia et al., 2024; Jiang et al., 2024; Wen et al., 2024). Across these settings, difficulty typically arises from the intrinsic complexity of the reasoning process, the length of the input or interaction, or the need to satisfy several clearly specified constraints simultaneously.

In this paper, we focus on a different – and largely unexplored – source of difficulty in instruction following. We study whether LLMs can follow instructions when they must autonomously determine which constraints are *applicable* to a specific task instance from a large pool of candidate constraints. While all constraints are valid and must be respected in general, only a small subset becomes binding in a given context. Unlike prior work, the instruction does not explicitly enumerate the applicable constraints. Instead, the model must (i) infer which constraints are contextually relevant and (ii) complete the task while satisfying those constraints. We refer to this setting as Large-Scale Constraint Generation (LSCG).

Figure 1 illustrates this challenge. The model is instructed to perform a social task (e.g., "be a good visitor in an Islamic country") and is provided with a comprehensive travel guide containing numerous generic behavioural recommendations. Although only a small subset of these recommendations is contextually binding, a naive approach that treats all guidelines as equally relevant can lead to inappropriate behaviour, such as inviting a Muslim for a beer after prayer (Naous et al., 2024). The core question in Large-Scale Constraint Generation (LSCG) is whether an LLM can infer, without additional instructions, which constraints are actually applicable in a given situation.

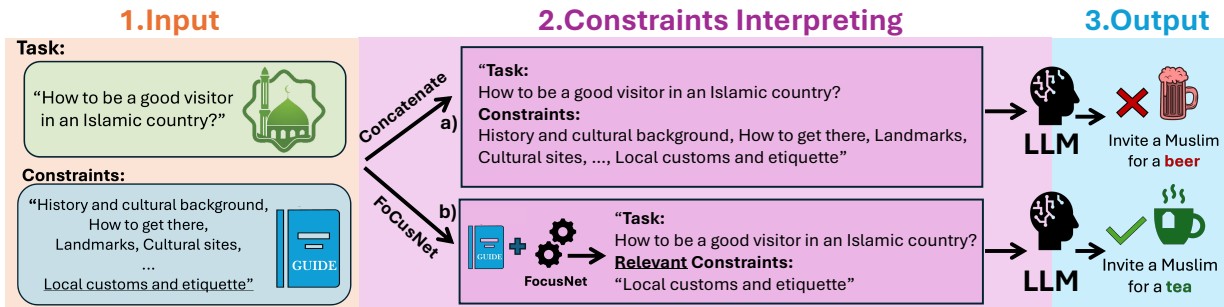

Figure 1: In LSCG, the model must generate a **valid answer** while adhering to an **input task** and a **long list of constraints**. In the example, this can be done either by (a) directly interpreting the **concatenated** task and constraints or (b) using a **FoCusNet** to **extract relevant constraints**. The first approach may lead to **inappropriate responses** (e.g., offering beer to a Muslim (Naous et al., 2024)), while the second ensures **valid answers**.

We argue that Large-Scale Constraint Generation (LSCG) reflects a fundamental requirement of contemporary LLM applications. Autonomous agents operating without direct human supervision are rarely provided with short, task-specific lists of constraints. Instead, they must retrieve and interpret relevant guidance from large collections of generic fine-grained rules, such as system specifications, security policies, or travel recommendations. Even when the underlying task is simple (e.g., "be a good visitor"), performance depends on the model's ability to identify which constraints are relevant, rather than complex logical reasoning.

This capability is more closely related to *practical intelligence* – the ability to navigate real-world situations by selecting and applying contextually appropriate rules – than to explicit *logical reasoning*, which emphasizes systematic deduction from clearly stated premises (Sternberg, 1986). As such, Large-Scale Constraint Generation (LSCG) isolates a dimension of instruction following that is orthogonal to task complexity and largely unaddressed by existing benchmarks.

To study this setting in a controlled manner, we introduce two concrete instances of Large-Scale Constraint Generation (LSCG): *Word Checker* and *Language Moderator*. In Word Checker, the model is given a long list of forbidden words and a sentence, and must classify the sentence as *valid* or *invalid*. The task is intentionally simple to isolate failures in constraint identification rather than reasoning. Language Moderator extends this setting by requiring the model to generate a semantically equivalent version of the sentence that avoids all forbidden words.

We create different task instances with increasingly larger lists of forbidden words (e.g., 100, 500 and 1000). Then, we systematically evaluate how features such as model family – Meta's *LLama* (Grattafiori et al., 2024) vs. Deepseek's *R1* (DeepSeek-AI et al., 2025)), size – 8B vs. 70B, and Test Steering Strategies (TSS)– *Simple Prompt*, *Chain of Thought* (Wei et al., 2022b; Lightman et al., 2024) and *Best of N* (Chen et al., 2024b; Madaan et al., 2023) affect the results. Finally, inspired by retrieval-augmented approaches (Lewis et al., 2020; Cobbe et al., 2021; Shi et al., 2024), we propose FoCusNet (Focused Constraints Net), a lightweight and customizable preprocessing model that reduces the effective constraint set by identifying candidates that are likely to be applicable to the task. In Word Checker, FoCusNet (Focused Constraints Net) is a ∼300k-parameter model trained to detect whether words appear in a sentence. At inference time, it filters the original constraint pool into a smaller set of potential suspects, enabling the LLM to focus on the constraints that matter.

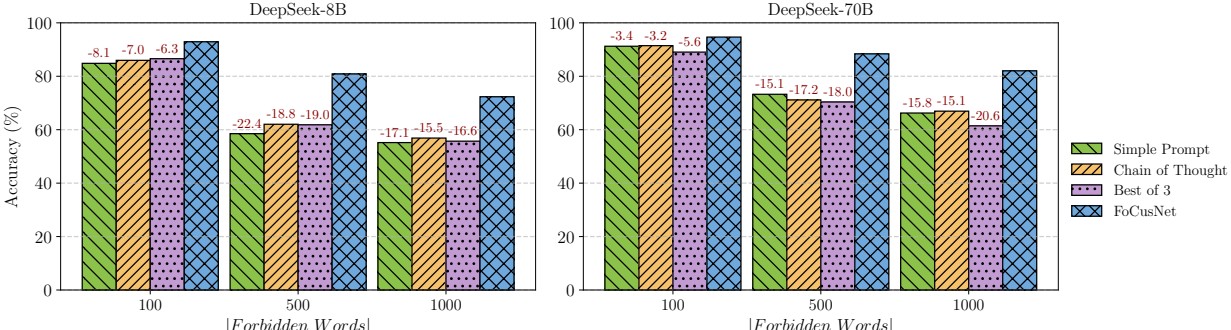

Figure 2: **FoCusNet consistently outperforms standard LLM inference strategies** (simple prompting, Chain-of-Thought, and Best-of-3) on the proposed Word Checker task, across both 8B and 70B models. While **baseline methods exhibit** substantial **accuracy degradation** as the number of forbidden words increases, **FoCusNet remains robust**. Red numbers indicate absolute accuracy differences relative to FoCusNet.

The results are striking. Figure 2 shows the performance in Words Checker for LLMs of 8B and 70B-parameters as the number of candidate constraints increases. Across model sizes, standard inference strategies – including simple prompting – exhibit a sharp degradation in accuracy, with drops of up to 30%. In particular, scaling the model alone does not mitigate this failure mode.

Manual analysis reveals that, under long constraint lists, LLMs frequently lose track of the task structure. Models often process candidate words in isolation, conflate intermediate reasoning with final answers, or incorrectly assert the presence of a word simply because it appears in a self-generated list rather than in the input sentence.

In contrast, our approach remains substantially more robust by decomposing the problem into two complementary models. FoCusNet, which detects the presence of forbidden words with approximately 90% accuracy, reduces the original constraint set to a small pool of probable candidates (on average, 30 words out of 1000). The LLM then operates on this narrowed scope, filtering out false positives from FoCusNet and enforcing the final decision. This division of labor preserves accuracy even under large pools of candidate constraints.

In summary, we make the following contributions.

- **Formulation of Large-Scale Constraint Generation**: We introduce a novel instruction-following setting in which LLMs must autonomously identify relevant constraints from a large pool of candidate constraints, most of which are irrelevant.

- **Instances of Large-Scale Constraint Generation**: We propose two concrete instances of LSCG– *Word Checker* and *Language Moderator* – which isolate the challenge of constraint identification under increasing constraint overload. Across both tasks, we systematically evaluate two model families (*LLaMA* and *R1*), two model sizes (8B and 70B), and three inference strategies (simple prompting, Chain-of-Thought, and Best-of-$N$).

- **FoCusNet**: We introduce FoCusNet, a lightweight auxiliary model that preprocesses long constraint lists by identifying likely relevant candidates, substantially improving LLM performance when used in conjunction with standard inference strategies.

- **Code and Datasets**: We release code and datasets for our tasks and FoCusNet to facilitate reproducibility and support future research on LSCG.[1]

---

[1] https://anonymous.4open.science/r/Large_Scale_Constraint_Generation-DDC2/

## 2 Related Work

**Instruction-Following Abilities of LLMs**. The challenge of constraining textual generation has been studied since the early days of NLP (Hu et al., 2017), but the rise of LLMs has dramatically increased expectations beyond merely "producing plausible text" (Brown et al., 2020; Wei et al., 2022a). Modern LLMs are now expected to follow complex instructions, satisfy multiple constraints across interactions (He et al., 2024c;b), and process long and structured inputs (Bai et al., 2024; Li et al., 2024).

Previous evaluations of instruction-following capabilities have yielded a rich set of insights. For instance, LLMs often struggle with strict rule adherence (Mu et al., 2024), exhibit a large variance in format-following performance across domains (Xia et al., 2024), lag behind closed-source models in complex instruction settings (Wang et al., 2023), and show pronounced weaknesses at smaller scales when faced with structured tasks (Wang et al., 2025). However, most of these benchmarks assume interactive, chat-like settings in which the model is given a small number of clearly specified, task-relevant instructions. In contrast, we study a setting in which the model is provided with an extensive list of fine-grained candidate constraints, only a small subset of which is relevant to the task at hand.

Closely related benchmarks such as *FollowBench* (Jiang et al., 2024) and *ComplexBench* (Wen et al., 2024) examine how instruction-following performance degrades as the number of constraints increases, by composing up to five and six constraints, respectively. However, in these benchmarks, all provided constraints are explicitly relevant and must be satisfied. Our work departs from this assumption by considering scenarios in which the constraint pool is orders of magnitude larger (hundreds to thousands), while only a few constraints – no more than four – are contextually applicable and capable of invalidating an otherwise correct response.

Finally, *CFBench* emphasizes the importance of evaluating instruction-following capabilities under realistic and authentic conditions. While our formulation of LSCG aligns with the requirements of real-world LLM applications such as autonomous agents, we intentionally focus on two simple task instantiations – Word Checker and Language Moderator – to isolate the impact of large candidate constraint pools from other sources of difficulty, such as complex reasoning or long-horizon planning.

**Instruction Tuning**. When dealing with instruction-following capabilities, *instruction tuning* might seem like a natural candidate to improve adherence to complex and fine-grained constraints. Previous work has highlighted its role in enhancing generalization capabilities (Chung et al., 2022; Mishra et al., 2022; Thoppilan et al., 2022), and even a small set of high-quality instructions can lead to performance gains (Zhou et al., 2023; Chen et al., 2024a). However, despite well-established guidelines for crafting such instructions (Zhao et al., 2024; He et al., 2024a; Zhang et al., 2024), instruction tuning remains costly and resource-intensive. This makes it unsuitable for large-scale applications that require customization (Chang et al., 2016; Zhang & Chen, 2020), continuous knowledge updates (Lewis et al., 2020), or, like our example in Fig. 2, cultural adaptation (Adilazuarda et al., 2024; Kotek et al., 2023). Instead, we argue that LLMs should, like humans, handle unfamiliar constraints by leveraging external knowledge sources while relying on their reasoning abilities to interpret and respond accordingly. Consequently, we do not employ instruction tuning to further specialize our models.

**Test Steering Strategies**. Instead of modifying a model through instruction tuning, an alternative approach is to guide LLM outputs at inference using test-time steering strategies. These methods improve adherence to the rules without the cost and inflexibility of fine-tuning. Previous research has explored various controlled generation techniques to enforce constraints (Hu et al., 2018). LLMs have shown strong performance with simple interventions like Chain-of-Thought (CoT) prompting (Wei et al., 2023). However, studies suggest that such methods alone may be insufficient for handling fine-grained, hard constraints (Sun et al., 2023). To address this, researchers have investigated best-of-$K$ selection (Nakano et al., 2022; Stiennon et al., 2020), where multiple independent samples are generated, scored, and ranked to select the most suitable output. Building on this body of work, we assess the rule-following capabilities of LLMs using various test-time steering strategies.

**Auxiliary Modules for LLMs**. In this article, we present FoCusNet, a modular support model that enhances the ability of LLMs to follow constraints. Unlike base model modifications, FoCusNet acts as an auxiliary module that identifies and prioritizes relevant constraints, guiding the LLM's generation process. It

provides an intermediate solution between resource-heavy instruction tuning and simpler test-time steering methods, which, while more efficient, may struggle with complex tasks.

Similar approaches using specialized support models for LLMs have been explored in various text generation tasks. For example, retrieval-augmented generation (RAG) (Lewis et al., 2020; Shi et al., 2024) improves LLM responses by incorporating external knowledge, while classifier-based safeguards promote responsible generation (Sharma et al., 2025). Furthermore, researchers have also developed classifier-based content moderation systems (Chi et al., 2024; Inan et al., 2023; Rebedea et al., 2023) and output filtering techniques to address jailbreak vulnerabilities (Kim et al., 2024).

**Classification as a Proxy for Constrained Generation**. We formulate Word Checker, one instance of LSCG, as a classification task that serves as a controlled proxy for constrained text generation. Using classification as a proxy for generation is common in benchmark design, motivated by the lack of reliable and standardized evaluation metrics for the generation of free text (Hendrycks et al., 2021; et al., 2022). Our goal is therefore not to assess fluency or creativity, but to evaluate whether LLMs can correctly reason about constraint satisfaction when faced with large pools of candidate constraints. As classification is generally easier than generation (Goodfellow et al., 2014), failure to detect constraint violations in a classification setting provides a *lower bound* on expected performance in constrained generation.

We complement Word Checker with Language Moderator, a more challenging generative task that additionally requires producing a semantically equivalent sentence while avoiding forbidden words. Together, these tasks enable a focused analysis of constraint overload while disentangling constraint identification from semantic reasoning and surface-level generation quality.

# 3 Large-Scale Constraint Generation

In this Section, we define Large-Scale Constraint Generation (LSCG), discuss Test Steering Strategies techniques and finally introduce FoCusNet.

## 3.1 Formal Definition

In constrained generation, an LLM autoregressively generates an output sequence $y$ given an input task $t$ and a set of constraints $c = \{c_1, c_2, \ldots, c_C\}$. We define LSCG as a setting of constrained generation characterized by a large pool of candidate constraints (i.e., $C \geq 100$). While all constraints in $c$ are valid in general, only an (unknown) subset $k \ll C$ is *contextually applicable*, or *binding*, to a given task instance. The model must therefore identify which constraints are binding in the current context and generate a valid output.

We suppose both $t$, and the constraints $c_i$ with $i \in C$ to be string-based. Although this assumption does not cover the most general case (see Sect. 6), it is sufficient to model real-world scenarios such as the travel guide of Sect. 1.

We define the LLM input *query*: $q = e(t) \parallel p(c)$, where $\parallel$ is the concatenation. Specifically, here $e$ and $p$ are *Test Steering Strategies* that can be applied to improve model performance: $e$ is a function that *enhance* the definition of the task, while $p$ helps *parsing* the constraints. We provide more details in the next section.

We represent the LLM as a function $f_\theta : q \to y$. This means that the LLM generates an answer $y$ as $y = f_\theta(q)$ according to its pre-trained weights $\theta$. A model-generated answer $y$ is valid for a given query $q$ if it correctly solves the task $t$ while adhering to the constraints $c$.

## 3.2 Existing Test Steering Strategies

Here, we list the most prominent TSS previously identified in the literature and examine how they apply in our formulation. We provide a summary in Tab 1.

**Simple Prompt**. As both $t$ and $c$ are text-based, a natural approach is to simply *concatenate* them: $q = t \parallel c_1 \parallel c_2 \parallel \cdots \parallel c_C$.

Table 1: Summary of how different steering solutions produces the final query $q = e(t) \parallel p(c)$.

| Test steering | Enhance - $e(t)$ | Parse - $p(c)$ |
|---|---|---|
| Simple Prompt | $t$ | $c_1 \parallel c_2 \parallel \cdots \parallel c_C$ |
| Chain of Thought | $t \parallel g$ | $c_1 \parallel c_2 \parallel \cdots \parallel c_C$ |
| Best of N | $t \parallel g$ | $y_1 \parallel y_2 \parallel \cdots \parallel y_N$ |
| Explicit Reasoning | $t \parallel g$ | $\hat{c}$ |
| FoCusNet | $t \parallel g$ | $f_\phi(c)$ |

**Chain of Thought (CoT)**. To enhance the reasoning capabilities of the LLM, we modify $t$ by appending a guide phrase $g$, such as "*Think step by step*": $q = t \parallel g \parallel c_1 \parallel c_2 \parallel \cdots \parallel c_C$.

**Best of N**. Finally, to improve the interpretation of the $C$ constraints, we can involve a panel of $N$ judges (e.g., independent runs of the model), each performing CoT reasoning independently, followed by a recap step to produce the final answer. Formally, let $y_n = f_{n,\theta}(t \parallel g \parallel c_1 \parallel c_2 \parallel \cdots \parallel c_C)$ denote the answer of the nth judge, where $n \in N$. Then, we can aggregate all the responses into a refined query: $q = t \parallel y_1 \parallel y_2 \parallel \cdots \parallel y_N$.

Finally, motivated by the observation that only $k \ll C$ constraints are binding for a given task instance, we introduce an explicit two-stage strategy that separates constraint identification from task execution:

**Explicit Reasoning**. In the first stage, an LLM processes the task $t$ and the full constraint set $c$ to identify a subset of binding constraints $\hat{c} \subseteq c$, with $|\hat{c}| \approx k$. In the second stage, a (possibly different) LLM solves the task using only the reduced constraint set:

$$\hat{c} = f_{\theta_1}(t \parallel c_1 \parallel \cdots \parallel c_C), \quad q = t \parallel \hat{c}_1 \parallel \hat{c}_2 \parallel \cdots \parallel \hat{c}_{|\hat{c}|}.$$

The final output is then generated from $q$.

### 3.3 FoCusNet

**Definition**. Following the Explicit Reasoning policy, we aim to approximate the (unknown) set of contextually binding constraints using a learned constraint parser. Concretely, we model the constraint parsing function as

$$p(c) : c \to \hat{c},$$

where $\hat{c} \subseteq c$ denotes a variable-size set of predicted binding constraints. To this end, we introduce a dedicated model, FoCusNet, defined as a function $f_\phi$ with learnable parameters $\phi$, trained on task-specific data to filter relevant constraints for a given task instance. Once trained, FoCusNet applies this filtering as $\hat{c} = f_\phi(c)$, yielding the final query formulation:

$$q = t \parallel g \parallel \hat{c}.$$

**Training FoCusNet**. We train FoCusNet to perform a binary classification task over individual constraints. Specifically, FoCusNet operates on triplets $(\hat{c}, s, l)$. Here, $\hat{c} = \{c_1, c_2, \ldots, c_M\}$ is a subset of $M$ constraints from $c$; $s$ is a text-based instance where the constraint is satisfied or violated, and $l \in \{0, 1\}$ is a label indicating whether the constraint is violated (1) or not (0). For example, consider Fig. 1. The set of constraints is $\{c_1 = $"*Respect local customs and etiquette when visiting an Islamic country*"$\}$; the instance is $s = $ "*Invite a Muslim for a beer*"; the corresponding label $l$ is violated ($l = 1$).

**Inference with FoCusNet**. During inference, FoCusNet receives as input the tuple of constraints and task $(c, t)$ and generates a *relevance mask*, $m = \{m_1, m_2, \ldots, m_C\}$ with $m_i \in \{0, 1\}$ and $i \in C$. The mask determines which constraints are relevant for the task. Applying the mask yields the reduced set: $\hat{c} = \{c_i \mid m_i = 1, \forall i \in C\}$.

As in any alerting system, FoCusNet aims at compromising *recall* and *precision*. Ideally, we would like FoCusNet to reduce the number of false positives, i.e., irrelevant constraints mistakenly included. In fact, a large number of false positives leads to a larger and noisy set $k$. At the same time, it is essential to minimize false negatives, as excluding relevant constraints could hinder the LLM's ability to generate valid outputs.

# 4  Methodology

In this section, we discuss the engineering of Words Checker and Language Moderator. Consequently, we describe FoCusNet's training.

## 4.1  Words Checker

**Problem Definition**. Words Checker is an instance of LSCG, where an LLM must classify a sentence as *valid* or *invalid* based on a dynamically provided list of forbidden words. Formally, given a sentence $S = (w_1, w_2, \ldots, w_n)$ and a set of forbidden words $F = \{w_{f1}, w_{f2}, \ldots, w_{fm}\}$, the model must determine whether $S$ contains any word morphologically related to an element of $F$. A sentence is classified as *invalid* if $\exists w_{fi} \in F$, $\exists w_j \in S$ such that $w_{fi}$ is a stem[2] or lemma[3] of $w_j$, and *valid* otherwise.

For example, given the sentence "The athlete skied a snowy mountain" and $F = \{ski\}$, the output should be *invalid*, since "skied" is a morphological variant of "ski". In contrast, for "The bathroom has recently been cleaned" and $F = \{restroom\}$, the output should be *valid*, as restroom is neither a lemma nor stem of "bathroom".

**Rationale behind Words Checker**. We explicitly design Words Checker to study the impact of an increasing number of forbidden words on LLM performance. Therefore, unlike other constrained generation problems, this task does not require complex reasoning. Instead, we engineer Words Checker as a simple problem that, as we will see, a simple string-matching algorithm combined with stemming and lemmatization could potentially solve. In summary, Words Checker serves as an *in vitro* study on LSCG. At the same time, Words Checker has practical applications. Consider a scenario where $S$ is an LLM-generated response $y$ in a conversation, and $F$ consists of words the user explicitly wants to avoid (e.g., when paraphrasing text, for secret keeping, etc.,).

**Testing Dataset**. To construct a dataset for Words Checker, we use the *CommonGen* (Lin et al., 2020) benchmark, originally designed for traditional constrained text generation. Each entry in CommonGen consists of a sentence and a variable-sized list of $W$ words that are morphologically present in it. For example, an entry may contain "The athlete skied a snowy mountain" with the corresponding words ["ski", "snow"]. We derive our dataset from two partitions of CommonGen, namely the *challenge train sample* and *challenge validation sample*[4]. For these partitions, $W$ ranges from 1 to 4. Given a pool size of candidate forbidden words $|F|$, we: i) construct a vocabulary from all CommonGen partitions and ii) iterate over the selected partitions to generate valid and invalid samples. To create an *invalid* example, we retain $W$ CommonGen words and randomly sample $|F| - W$ additional vocabulary words. For a *valid* example, we select $|F|$ random words ensuring that none are a lemma or stem of the words in the sentence.

We generate four versions of Words Checker, each containing 1000 sentences, with increasing constraint complexity: $F = \{10, 100, 500, 1000\}$. We generate balanced datasets, with approximately equal support for both classes. Notice that the 1000 sentences are the same across all scenarios.

## 4.2  Language Moderator

**Problem Definition**. Language Moderator is a second instance of LSCG in which, similarly to Words Checker, the model receives as input a sentence and a set of forbidden words. In this setting, the model is asked to *moderate* the input sentence by removing any forbidden words, if present. This requires generating a new sentence that preserves the semantics of the original while eliminating all forbidden words.

For example, given the sentence "The athlete skied a snowy mountain" and $F = \{ski\}$, we expect the model to produce an alternative such as "The athlete slid down a snowy mountain." Conversely, given the sentence "The bathroom has recently been cleaned" and $F = \{restroom\}$, the model should leave the sentence unchanged.

---

[2]Word part responsible for a word's lexical meaning (e.g., "walk" → "walked"
[3]Canonical form, dictionary form, or citation form of a set of word forms (e.g., "break" → "broke"
[4]The test partitions of CommonGen do not contain reference sentences.

**Rationale behind Language Moderator**. As in Words Checker, Language Moderator enables a systematic study of LLM performance under increasing candidate constraint pools. Unlike Words Checker, however, it also requires generating a semantically equivalent sentence that satisfies all binding constraints. This introduces a minimal but non-trivial reasoning component on top of constraint identification.

We show that under large-scale constrained generation, this additional requirement leads to a drastic performance degradation. Although the semantic transformation itself is simple, the burden of identifying binding constraints among a large pool interferes with the LLMs' ability to perform even basic semantic reasoning. As we will show, by mitigating large-scale constrained generation through explicit constraint parsing, FoCusNet alleviates this cognitive overload and restores the model's ability to perform the underlying reasoning task. This demonstrates that the observed failures stem not from limitations in semantic reasoning per se, but from the difficulty of identifying and managing relevant constraints at scale.

**Testing Dataset**. Language Moderator uses exactly the same data instances as Word Checker. In particular, we reuse the same sentences, the same pools of candidate forbidden words $F$, and the same valid and invalid assignments derived from the CommonGen benchmark (Lin et al., 2020). Language Moderator requires generating a moderated version of the sentence: invalid instances must be rewritten to remove all forbidden words while preserving the original semantics, whereas valid instances should remain unchanged.

### 4.3 FocusNet for Word Checker and Language Moderator

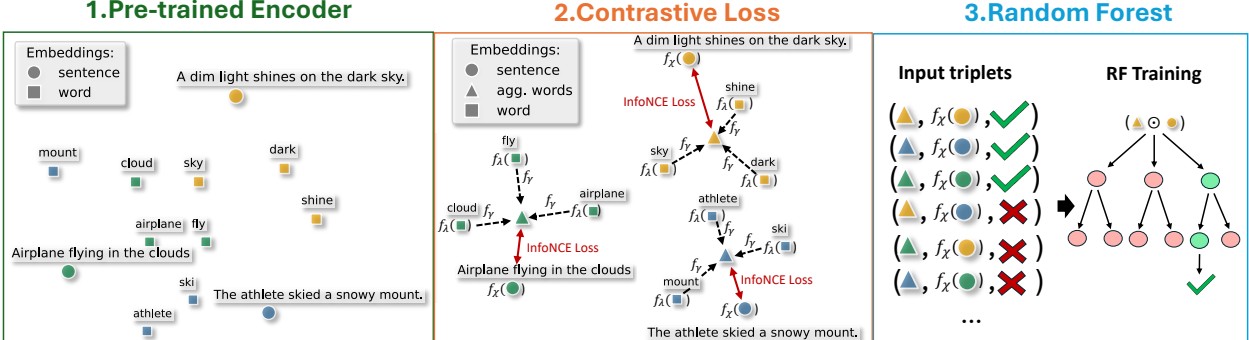

Figure 3: Training pipeline of FoCusNet. The model receives as input a batch of sentences and words. In Phase 1, FoCusNet uses a **frozen pre-trained model** to map the input into sentences (circles) and words (squares) embeddings. Then, in Phase 2, FoCusNet learns to **refine the sentence embeddings** ($f_\chi$) and to **aggregate the words embeddings** ($f_\gamma$, $f_\lambda$) with a InfoNCE contrastive loss. Eventually, in Phase 3 FoCusNet train a Random Forest to **discriminate positive and negative examples**.

**Model Description**. To tackle Word Checker and Language Moderator, we train FoCusNet to recognize whether a sentence $S$ contains a set of words $W = \{w_1, w_2, \ldots, w_n\}$. The training pipeline, summarised in Fig. 3, is divided into three phases:

*Phase 1*: We use a frozen pre-trained sentence encoder to obtain the initial embeddings for the sentence ($e_S$) and the words ($\{e_{w_1}, e_{w_2}, \ldots, e_{w_n}\}$).

*Phase 2* Next, we refine these embeddings through two learnable projection layers. The sentence embeddings are refined with a linear layer $f_\chi : e_S \to \hat{e}_S$, where $\hat{e}_S$ is the refined sentence embedding. We aggregate the word embeddings into a single refined embedding $e_{\hat{w}}$ using an attention mechanism (Bahdanau, 2014). Specifically, given the embeddings $e_{w_1}, e_{w_2}, \ldots, e_{w_N}$, we compute $e_{\hat{w}}$ as:

$$e_{\hat{w}} = \sum_{i=1}^{N} f_\gamma(e_{w_i}) \cdot f_\lambda(e_{w_i})$$

Intuitively, we use this aggregation layer and focus on more words simultaneously to give the model a broader understanding of the context in which the words are used. For example, with $\{W_1 = $ "mount", "ski"$\}$ and $W_2 = \{$"mount", "lake"$\}$, the model understands that "mount" refers to both human activity / infrastructure and natural geography / environment.

We train the layers $\chi$, $\gamma$, and $\lambda$ using the *InfoNCE* loss (Oord et al., 2018), which encourages higher cosine similarities for sentences and words that appear in the same set $W$. Specifically, two sentences $S_1$ and $S_2$ from the same batch are considered positive examples if they share the same set of words, and negative otherwise.

*Phase 3*: After training the encoder and projection layers, we concatenate the refined sentence embedding $\hat{e}_S$ and the word embedding $e_{\hat{w}}$ into a final embedding $e_f = \hat{e}_S \parallel e_{\hat{w}}$. This concatenated embedding is then fed into a Random Forest classifier, which determines whether the words encoded in $e_{\hat{w}}$ appear in the sentence $S$ or not.

The last two phases of the training pipeline draw inspiration from the *Supervised Contrastive Loss* paper (Khosla et al., 2020), and are designed to learn high-quality embeddings.

**Training Dataset**. To train FoCusNet, we use the remaining *train* and *validation* splits of the CommonGen dataset. Since more than 80% of sentences contain exactly three target words, we apply synthetic data augmentation to increase variability in the number of words contained.

Given a sentence (e.g., "The athlete skied a snowy mountain") associated with three target words (e.g., "athlete", "ski", "mountain"), we randomly sample subsets of size one (e.g., "mountain") or two (e.g., "athlete", "ski"). The original sentence is treated as a valid positive example for each sampled subset. This augmentation strategy enables the model to learn from training examples with varying numbers of target words, thereby improving its ability to generalize.

Finally, to further improve robustness and preserve semantic grounding, we augment the dataset with (word,definition) pairs from WordNet, the lexical database used by the Python NLP library `nltk`[5]. For each word appearing in CommonGen, we retrieve its WordNet definition and treat it as a positive example for the contrastive loss. This procedure adds approximately 6k additional samples to the dataset.

We refer to the resulting merged dataset as *CommonNet*. In total, CommonNet contains approximately 220k labeled examples consisting of sentences and their associated target words.

## 5 Experiments

This section presents a structured evaluation of Large-Scale Constraint Generation (LSCG). We first establish the difficulty of the problem by fixing a Simple Prompt strategy and measuring how performance degrades as the number of candidate constraints increases, showing that LLMs struggle with constraint overload even in a minimal setting. We then compare standard test-time steering strategies (Chain-of-Thought and Best-of-N) against FoCusNet on the Word Checker task, analyzing the effects of model family, model size, and constraint pool size. Next, we conduct a fine-grained analysis of the parsing skills of LLMs when assisted by FoCusNet, evaluating whether models correctly retrieve the specific forbidden words responsible for invalid classifications. Finally, we extend the study to Language Moderator, a constrained generation task that additionally requires semantic preservation, to assess how large-scale constraint overload impacts generation and whether explicit constraint parsing mitigates this effect.

### 5.1 Experiments Settings

**LLMs Inference**. To deploy the LLMs in our Words Checker experiments, we use $SGLang$[6], an open-source framework that facilitates efficient model downloading and deployment. Specifically, we select four models from SGLang's library: *Meta-Llama-3.3-8B-Instruct* and *Meta-Llama-3.3-70B-Instruct* from the LLaMA family (Grattafiori et al., 2024), as well as the more recent *DeepSeek-R1-Distill-Llama-8B* and *DeepSeek-R1-*

---

[5] `https://www.nltk.org/`
[6] `https://docs.sglang.ai/index.html`

*Distill-Llama-70B* from DeepSeek (DeepSeek-AI et al., 2025). The deployment of the 70B models required four NVIDIA RTX A6000 GPUs, whereas the 8B models ran efficiently on a single A6000 GPU. When prompting the models, we set the *temperature t* to 0.2 for the Simple Prompt strategy and increase it to 0.4 for more sophisticated TSS. The exact prompts used are provided in Appendix A. We set N=3 when using the Best of N strategy.

**Training FoCusNet**. For the contrastive loss training of FoCusNet, we perform a hyperparameter search using 4-fold cross-validation ($K = 4$), ensuring that all examples sharing the same word list are assigned to the same fold to prevent data leakage. We explore embedding sizes $\{64, 128, 256, 512\}$, learning rates $\{1e^{-4}, 2.5e^{-4}, 5e^{-4}\}$, and InfoNCE loss temperatures $\{0.05, 0.1, 0.2\}$, training for 30 epochs. The best configuration, determined by averaging validation results, consists of an embedding size of 128, a learning rate of $2.5e^{-4}$, a temperature of 0.05, and 24 training epochs, using *all-mpnet-base-v2*[7] as the pre-trained encoder. After selecting the best encoder, we train a random forest where each sentence is paired with a positive (words contained in the sentence) and a negative example (words not contained). A hyperparameter search yields an optimal configuration of 200 trees, a maximum depth of 10, and a minimum of 3 samples per leaf. We report an ablation study of FocusNet' design choices in Appendix A.

**Metrics**. For Words Checker, we evaluate performance using *accuracy* (overall correctness), *precision* (the proportion of predicted positive sentences that actually contain at least one forbidden word), and *recall* (the proportion of actual positive sentences correctly identified). Additionally, for invalid sentences, we assess the LLMs' parsing ability when relying on FoCusNet. For invalid sentences, we ask the LLM to retrieve invalid words from the sublist FoCusNet already pre-processed and measure its *parsing precision* and *parsing recall*. For example, given the sentence "The athlete skied the snowy mountain," the set of forbidden words {snow, mountain, ski}, and the LLM's prediction {snow, ski, sun, fun}, the parsing recall is 0.66 (2 out of 3 correct words retrieved), while the parsing precision is 0.5 (2 out of 4 predicted words are correct).

For Language Moderator, we evaluate how often the LLM correctly detects a policy violation in the original sentence and produces a revised, compliant version (a *true positive*). When a correction is generated, we first verify that the revised sentence is violation-free using a rule-based string checker with stemming and lemmatization. To be considered violation-free, the model must resolve all violations present in the original sentence. The dataset contains, on average, 1.73 violations per sentence. We then assess whether the revised sentence preserves the meaning of the original using an *LLM-as-a-judge* approach. Specifically, we use *gpt-4.1* as a judge and implement the evaluation with the Python library *deepeval*[8]. We set the acceptance threshold to 0.8. The evaluation prompt is reported in Appendix A.

## 5.2 Results

**Is Words Checker challenging?**. We assess the effectiveness of a simple prompting strategy and find that all models, regardless of family or size, experience a roughly 30% accuracy drop as the number of forbidden words increases from 10 to 1000 (see Fig. 4). In addition, more forbidden words lead to an increase in false alarms. For example, with 100 forbidden words, LLama 70B has a recall of 97% and precision of 99%, but with 1000 forbidden words, the recall only decreases to 92%, while the precision drops to 65%. These results show that, despite simplicity, Words Checker remains challenging for basic prompting strategies, suggesting that more advanced Test Steering Strategies are needed.

**FoCusNet vs. Traditional Test Steering Strategies**. We assess the impact of advanced Test Steering Strategies, like *Chain of Thought*, *Best of 3*, and *Explicit Reasoning* on Words Checker using Deepseek's R1-8B and Deepseek's R1-70B models and compare the results with FoCusNet. We also include a simple *RAG baseline* in which a pre-trained encoder retrieves the top-30[9] forbidden words given the sentence.

Observe the results in Tab. 2. For $|F| = 100$, all strategies achieve high accuracy across both DeepSeek-8B and DeepSeek-70B, indicating that the task remains relatively simple. Differences arise mainly in the precision–recall trade-off: RAG achieves very high recall but suffers from lower precision, while traditional

---

[7] https://huggingface.co/sentence-transformers/all-mpnet-base-v2
[8] https://deepeval.com/
[9] 30 is the average length of candidates retrieved by FoCusNet

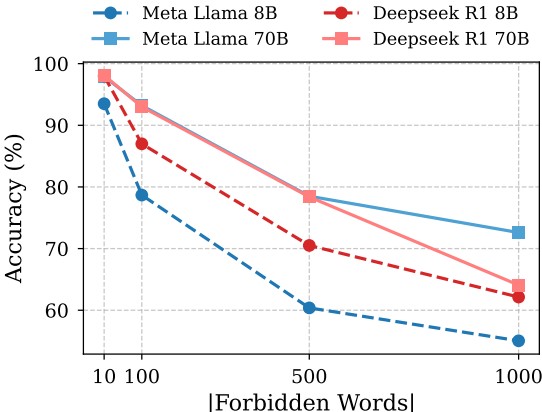

Figure 4: Accuracies with a "Simple Prompt" strategy as the number of forbidden words increases.

Table 2: Results of DeepSeek-R1-Distill models under different Test Steering Strategies as the number of forbidden words $|F|$ increases. The proposed FoCusNet consistently outperforms other TSS methods.

| Model | Strategy | $|F| = 100$ | | | $|F| = 500$ | | | $|F| = 1000$ | | |
|---|---|---|---|---|---|---|---|---|---|---|
| | | Acc. | Rec. | Prec. | Acc. | Rec. | Prec. | Acc. | Rec. | Prec. |
| DeepSeek-8B | Simple Prompt | 84.84 | 96.52 | 80.43 | 58.51 | 93.86 | 56.58 | 55.18 | 91.53 | 54.44 |
| | Chain of Thought | 85.96 | 94.84 | 82.46 | 62.04 | 94.36 | 59.48 | 56.87 | 88.21 | 55.76 |
| | Best of 3 | 86.57 | 95.73 | 80.87 | 61.83 | 90.41 | 58.35 | 55.71 | 88.73 | 53.48 |
| | Elicit Reasoning | 62.39 | 94.94 | 63.34 | 52.71 | 95.09 | 52.56 | 51.26 | 97.08 | 51.15 |
| | RAG | 92.04 | **98.56** | 86.96 | 79.74 | **96.93** | 71.78 | 70.93 | **97.13** | 63.24 |
| | **FoCusNet** | **92.91** | 96.71 | **89.69** | **80.87** | 95.97 | **74.26** | **72.33** | 95.76 | **66.50** |
| DeepSeek-70B | Simple Prompt | 91.24 | 91.17 | 91.36 | 73.26 | 87.47 | 70.07 | 66.24 | 90.07 | 61.57 |
| | Chain of Thought | 91.45 | 91.25 | 91.63 | 71.14 | 88.86 | 66.85 | 66.94 | 88.25 | 62.48 |
| | Best of 3 | 89.05 | 91.16 | 87.47 | 70.37 | 88.13 | 65.42 | 61.44 | 90.35 | 57.62 |
| | Elicit Reasoning | 62.04 | **95.45** | 62.60 | 53.11 | **96.32** | 53.29 | 51.22 | **97.21** | 51.48 |
| | RAG | 94.20 | 94.47 | 93.70 | 87.96 | 94.44 | 83.15 | 77.90 | 91.24 | 72.13 |
| | **FoCusNet** | 94.67 | 91.74 | **97.16** | **88.38** | 92.29 | **85.52** | **82.03** | 94.44 | **75.12** |

prompting and reasoning-based strategies perform similarly. In this setting, FoCusNet already achieves the best or near-best accuracy, driven by a clear improvement in precision without compromising recall.

When increasing the forbidden set to $|F| = 500$, the weaknesses of traditional test steering strategies become apparent. Although recall remains high for most methods – particularly RAG and Elicit Reasoning – precision drops significantly, leading to notable accuracy degradation. Chain-of-Thought and Best-of-3 often underperform Simple Prompt, suggesting that forcing additional reasoning in a largely mechanical task induces overthinking, confusion, and hallucinated forbidden terms (see Appendix A). In contrast, FoCusNet consistently outperforms all baselines for both model sizes, achieving the highest accuracy and precision while maintaining strong recall. By focusing the model on a small relevant subset of forbidden words, FoCusNet reduces false positives and keeps the model aligned with the task.

Finally, at $|F| = 1000$, these issues are amplified. Traditional TSS approaches approach chance-level accuracy despite high recall, reflecting an increasingly severe precision collapse. While RAG remains recall-oriented, its accuracy deteriorates as the forbidden set grows. FoCusNet, instead, degrades gracefully: it maintains substantially higher precision and accuracy than all competitors, for both DeepSeek-8B and DeepSeek-70B. Notably, FoCusNet allows the 8B model to match or exceed the performance of traditional strategies applied to the 70B model, highlighting the effectiveness of focused test-time steering over model scaling.

Overall, FoCusNet consistently delivers superior performance across all difficulty regimes, achieving a better balance between recall and precision and demonstrating robustness as the number of forbidden words increases.

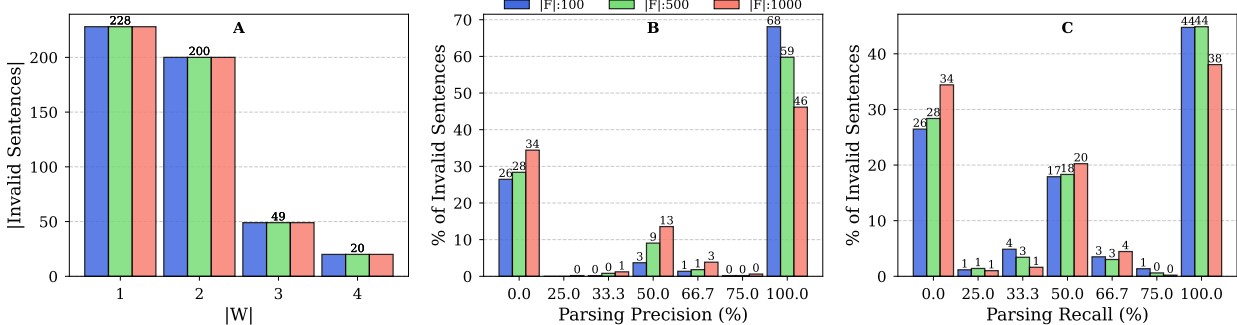

Figure 5: Figure A shows the distribution of the number of forbidden words per invalid sentence. Figures B and C report the distributions of parsing precision and recall, respectively, computed per sentence. In all cases, the LLM operates on the list of candidate forbidden words produced by FoCusNet.

**Parsing Skills of the LLM When Using FoCusNet**.

We now conduct a more fine-grained evaluation by analysing the LLM's ability to *extract* forbidden words when assisted by FoCusNet. In this setting, rather than only classifying a sentence as valid or invalid, the LLM must explicitly output the forbidden words it detects in the sentence. Concretely, FoCusNet provides the LLM with a set of candidate forbidden words for each sentence, and the LLM returns a subset of these candidates as its prediction.

For each invalid sentence, we compute *parsing precision* as the fraction of forbidden words predicted by the LLM that actually appear in the sentence, and *parsing recall* as the fraction of true forbidden words that are correctly identified by the LLM. Both metrics are computed on a per-sentence basis. We use Deepseek's R1-8B as the base model.

Our analysis focuses on the *497 invalid sentences* in the dataset, i.e., all sentences that contain at least one forbidden word ($|W| \geq 1$). The results are summarized in Fig. 5. Subfigure A reports the distribution of the number of forbidden words per invalid sentence. As shown, most invalid sentences contain either one or two forbidden words.

Subfigures B and C plot the distributions of parsing precision and parsing recall, respectively. The *x-axis* reports discrete precision/recall values, while the *y-axis* shows the percentage of invalid sentences that achieve each value. For example, in subfigure B, when starting from the list of candidate words returned by FoCusNet, the LLM attains a parsing precision of 100% on 68% of invalid sentences. Both parsing precision and recall exhibit a clear trimodal distribution, with peaks at 0%, 50%, and 100%. This behavior directly follows from the dataset composition in subfigure A: since sentences typically contain one or two forbidden words, precision and recall can take only a small set of discrete values depending on whether the LLM identifies none, one, or all of them.

Although the proportion of perfect predictions (i.e., 100% precision or recall) consistently exceeds that of completely incorrect predictions (0%), increasing the number of candidate forbidden words ($|F|$) negatively impacts performance. Importantly, the experimental settings with $|F| = 100$ and $|F| = 500$ use the same set of invalid sentences; therefore, the sets of true forbidden-words $W$ are identical across these settings, and FoCusNet returns the same words relevant to ground-truth in both cases. However, as $|F|$ grows, FoCusNet is more likely to introduce false positives into the candidate set passed to the LLM. These spurious candidates increase ambiguity and distract the model, leading to additional errors in word identification, and consequently to lower parsing precision and recall.

**Language Moderator**. To conclude, we report the results of the Language Moderator task. We focus on two Steering Strategies (the simplest, Simple Prompt; the one with best results on Words Checker, FoCusNet) and use models from the Deepseek family (Deepseek's R1-8B and Deepseek's R1-70B). We report results under increasing forbidden-word set sizes ($|F| = 100, 500, 1000$).

Table 3: Results of DeepSeek-R1-Distill models for Language Moderator using Simple Prompt and FoCusNet. Indented rows correspond to progressively stricter success criteria: detecting a violation (True Positives), producing a violation-free rewrite (Infraction-Free), and finally preserving the original semantics (Semantically Valid).

| Model | Metric | Simple Prompt | | | FoCusNet | | |
|-------|--------|---------------|--------------|--------------|--------------|--------------|--------------|
| | | $\|F\| = 100$ | $\|F\| = 500$ | $\|F\| = 1000$ | $\|F\| = 100$ | $\|F\| = 500$ | $\|F\| = 1000$ |
| **DeepSeek 8B** | **Support Invalid** | 488 | 488 | 488 | 488 | 488 | 488 |
| | **True Positives** | 463 (94.9%) | 334 (68.4%) | 304 (62.3%) | 481 (98.6%) | 464 (95.1%) | 444 (91.0%) |
| | **Infraction-Free** | 290 (59.4%) | 120 (24.6%) | 81 (16.6%) | 360 (73.8%) | 248 (50.8%) | 154 (31.6%) |
| | **Semantically Valid** | 160 (32.8%) | 37 (07.6%) | 19 (03.9%) | 208 (42.6%) | 105 (21.5%) | 58 (11.9%) |
| | **Support Valid** | 512 | 512 | 512 | 512 | 512 | 512 |
| | **True Negatives** | 316 (61.7%) | 294 (57.4%) | 258 (50.4%) | 325 (63.5%) | 239 (46.7%) | 215 (42.0%) |
| **DeepSeek 70B** | **Support Invalid** | 488 | 488 | 488 | 488 | 488 | 488 |
| | **True Positives** | 486 (99.6%) | 460 (94.3%) | 436 (89.3%) | 482 (98.8%) | 480 (98.4%) | 480 (98.4%) |
| | **Infraction-Free** | 411 (84.2%) | 252 (51.6%) | 134 (27.5%) | 404 (82.8%) | 307 (63.0%) | 241 (49.5%) |
| | **Semantically Valid** | 302 (61.9%) | 173 (35.5%) | 87 (17.8%) | 294 (60.2%) | 217 (44.5%) | 166 (34.0%) |
| | **Support Valid** | 512 | 512 | 512 | 512 | 512 | 512 |
| | **True Negatives** | 370 (72.3%) | 306 (59.8%) | 286 (55.9%) | 334 (65.2%) | 278 (54.3%) | 215 (42.0%) |

Observe the results of Tab. 3. Across both model sizes, Simple Prompt exhibits a pronounced degradation as $|F|$ increases, particularly for the stricter metrics. While violation detection remains strong for small constraint sets (e.g., 94.9% true positives for DeepSeek-8B and 99.6% for DeepSeek-70B at $|F| = 100$), performance drops substantially as the constraint pool grows. A large fraction of this drop already occurs between *True Positives* and *Infraction-Free* outputs, indicating that models frequently identify the presence of a violation but fail to remove all forbidden terms, resulting in revised sentences that remain invalid. This effect becomes more pronounced as $|F|$ increases, reflecting the difficulty of simultaneously tracking and correcting multiple binding constraints.

The degradation is further amplified when semantic preservation is required. For DeepSeek-8B, semantically valid corrections fall from 32.8% at $|F| = 100$ to 3.9% at $|F| = 1000$; similarly, DeepSeek-70B drops from 61.9% to 17.8%. These results indicate that even when the model detects a violation, it increasingly fails to generate a meaning-preserving rewrite under large-scale constraints.

In contrast, FoCusNet substantially mitigates this degradation, maintaining high true-positive rates even at $|F| = 1000$ (e.g., above 91% for DeepSeek-8B and above 98% for DeepSeek-70B). More importantly, FoCusNet improves constraint compliance: the gap between *True Positives* and *Infraction-Free* outputs is consistently reduced, indicating that explicit constraint parsing helps models identify and eliminate all relevant forbidden terms. As a consequence, semantically valid rewrites increase markedly across all settings, with the largest gains occurring for larger constraint pools (e.g., from 17.8% to 34.0% for DeepSeek-70B at $|F| = 1000$).

However, Table 3 also reveals a systematic trade-off. In valid inputs, FoCusNet tends to be more conservative, leading to fewer true negatives than Simple Prompt in several settings, particularly as $|F|$ grows. This behavior suggests that while explicit constraint parsing improves recall over violations, it can also increase false positives, pointing to a precision–recall trade-off that warrants further investigation.

Overall, Table 3 confirms that Language Moderator is significantly more demanding than pure constraint detection, and that the addition of even simple semantic rewriting exacerbates performance collapse under large-scale constraints. By alleviating constraint-identification overload, FoCusNet enables models to recover both detection accuracy and semantic fidelity, validating the rationale behind its design.

## 6 Conclusions

We introduced *Large-Scale Constraint Generation* (LSCG), a new instruction-following setting in which large language models must identify which constraints are relevant to a task from a large pool of candidates, most of which are contextually irrelevant. Unlike prior work that assumes a small number of explicitly binding

constraints, LSCG isolates a distinct and under-explored failure mode arising from constraint relevance identification rather than from complex reasoning or long-context processing.

We studied LSCG through two tasks: *Word Checker*, a controlled classification setting designed to isolate constraint identification under increasing constraint overload, and *Language Moderator*, which extends this challenge to constrained generation by requiring meaning-preserving rewrites. Across model families, sizes, and inference strategies, we observe a sharp and consistent performance degradation as the number of candidate constraints increases. Model scaling and standard test-time steering methods, including Chain-of-Thought and Best-of-$N$, do not resolve this failure and often exacerbate false positives through overthinking and hallucinated constraints.

To mitigate this issue, we proposed *FoCusNet*, a lightweight auxiliary model that explicitly separates constraint identification from task execution by filtering large constraint pools to a small set of likely relevant candidates. FoCusNet consistently improves performance across both tasks, remains robust under large constraint sets, and enables smaller models to rival or outperform much larger ones without explicit constraint parsing.

Our findings highlight the identification of constraint relevance as a fundamental bottleneck for instruction-following systems and a critical capability for real-world applications such as autonomous agents, policy compliance, and safety moderation. By formalizing LSCG and providing open benchmarks and models, we aim to encourage further research on modular, constraint-aware architectures and more realistic evaluations of LLM behavior.

## Limitations

We outline the main limitations of the present work.

First, while we motivate Large-Scale Constraint Generation (Large-Scale Constraint Generation) with a wide range of real-world scenarios, our empirical study focuses on two specific task instantiations: Words Checker and Language Moderator. These tasks are intentionally designed to minimize the role of complex semantic reasoning, long-horizon planning, or linguistic creativity, to isolate a single factor: the identification of contextually relevant constraints from large candidate pools. As a result, our analysis does not cover more complex settings in which constraint relevance interacts with deeper reasoning, multi-step decision making, or open-ended generation. Extending LSCG to richer task families remains an important direction for future work.

Second, our proposed auxiliary model, FoCusNet, relies on the availability of task-specific supervision to learn effective constraint filtering. This dependence may limit applicability in domains where labeled data or reliable synthetic augmentation strategies are scarce. Although we argue that FoCusNet does not require a strong out-of-distribution generalization – since its role is to conservatively filter candidate constraints rather than solve the task itself – future work should more systematically explore the trade-off between data availability, filtering accuracy, and downstream LLM performance across diverse LSCG settings.

Third, although we present FoCusNet as a generic modular extension for LLMs, its architecture and training procedure have only been evaluated in the context of textual constraints of word-level. It remains unclear how well the same design would transfer to qualitatively different constraint types, such as structural rules, logical dependencies, or abstract policies. Investigating alternative architectures and training signals for constraint parsers is an important avenue for future research.

Finally, our study is limited to purely textual constraints. However, in many realistic applications, constraints can span multiple modalities, such as vision, audio, or structured data (Chi et al., 2024; Inan et al., 2023). Identification of Large-scale constraint relevance in multimodal settings poses additional challenges, including representation alignment and cross-modal grounding. Adapting the LSCG framework and auxiliary filtering mechanisms such as FoCusNet to such settings is a promising direction for future work.

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

# A  Appendix

## A.1  Ablation Study on FoCusNet

Here we report an ablation study conducted on FoCusNet (FoCusNet), aimed at understanding the contribution of its main architectural and training components.

Specifically, we investigate: (i) whether further training a pre-trained encoder with a contrastive loss is necessary, or if off-the-shelf sentence embeddings are sufficient; (ii) whether aggregating word embeddings via an attention mechanism (as shown in Fig. 3) is beneficial, compared to training the contrastive module on one word at a time; (iii) the importance of semantic supervision by removing the WordNet component from training and relying solely on CommonGen; and (iv) the impact of the classification head by replacing the Random Forest with a simpler KNN classifier. Finally, we compare FoCusNet against a purely symbolic baseline based on lemmatization and stemming.

In this ablation, we focus on two complementary aspects:

- The ability of FoCusNet to identify whether a sentence contains forbidden words.

- The ability of FoCusNet to preserve semantic grounding in the learned embedding space.

**Forbidden-word detection.**  To evaluate the first aspect, we select a subset of sentences from the CommonGen dataset that was never used during training. This subset contains approximately 7k sentences. From these, we construct a balanced evaluation set of 14k samples by pairing each sentence with either (i) a lemma or stem that appears in the sentence (positive samples, which should be classified as invalid), or (ii) a lemma or stem that does not appear in the sentence (negative samples).

The results are reported in Table 4.

Table 4: Ablation study of FoCusNet when identifying whether a sentence contains an invalid stem or lemma.

| Contrastive Learning | Attention | Training Corpus | Classifier | Accuracy | Recall | Precision | F1 |
|---|---|---|---|---|---|---|---|
| Yes | Yes | CommonNet | Random Forest | 0.86 | 0.95 | 0.80 | 0.87 |
| Yes | No | CommonNet | Random Forest | 0.85 | 1.00 | 0.76 | 0.87 |
| Yes | Yes | CommonGen | Random Forest | 0.84 | 0.89 | 0.81 | 0.85 |
| Yes | Yes | CommonNet | KNN | 0.87 | 0.93 | 0.83 | 0.87 |
| No | No | CommonNet | Random Forest | 0.79 | 0.73 | 0.82 | 0.77 |
| No | No | None | None | **1.00** | **1.00** | **1.00** | **1.00** |

*Note: The first row corresponds to the default configuration of the system.*

First, removing contrastive learning and relying solely on a frozen encoder results in a clear performance degradation (fifth row), indicating that off-the-shelf sentence embeddings are insufficient for this task. Contrastive training is therefore crucial for aligning sentence and word representations in a way that supports reliable constraint detection.

Second, the attention-based aggregation of word embeddings has a moderate but consistent impact. While disabling attention does not drastically affect accuracy, it increases recall at the expense of precision, suggesting that processing words independently leads to more conservative predictions and a higher false-positive rate. Attention helps the model jointly reason over multiple candidate words, improving the precision–recall trade-off.

Third, training on the combined CommonNet corpus yields better results than training on CommonGen alone. In particular, removing the WordNet-derived (word, definition) pairs degrades recall and F1 score, confirming that explicit semantic supervision improves robustness to morphological and lexical variation.

Fourth, replacing the Random Forest classifier with a simpler KNN produces comparable performance, indicating that most of the discriminative power lies in the learned embedding space rather than in the complexity of the classification head.

Finally, the rule-based string checker achieves perfect performance by construction, as it directly implements the evaluation rule. However, this baseline does not generalize beyond exact lexical matching and cannot capture semantic relatedness or contextual usage. Its inclusion therefore serves as an upper bound for purely symbolic methods rather than as a practical alternative to FoCusNet in realistic settings.

**Semantic grounding.** To assess the second aspect, we evaluate the semantic quality of the learned embedding space using a retrieval-based metric. We consider two subsets of words: 681 generic words and 1953 more specific words. For generic words, we pair each word with the definition of its root form as provided by WordNet; for specific words, we pair each word with its own definition.

For each sample, we compare the embedding of the definition against the embedding of the target word and an additional pool of 99 randomly sampled candidate words (100 candidates in total). We then compute the Mean Retrieval Index (MRI), defined as the fraction of cases in which the correct word appears among the top-$k$ nearest neighbors. Higher MRI values indicate better preservation of semantic structure.

The results are reported in Table 5.

Table 5: Generic and Specific MRI scores under different training configurations.

| Contrastive Learning | Attention | Training Corpus | Gen. $MRI_1$ | Gen. $MRI_5$ | Spec. $MRI_1$ | Spec. $MRI_5$ |
|---|---|---|---|---|---|---|
| Yes | Yes | CommonNet | 0.32 | 0.62 | 0.54 | 0.76 |
| Yes | No | CommonNet | 0.34 | 0.61 | 0.56 | 0.77 |
| Yes | Yes | CommonGen | 0.20 | 0.42 | 0.38 | 0.61 |
| Yes | Yes | CommonNet | 0.31 | 0.62 | 0.54 | 0.76 |
| No | No | CommonNet | 0.01 | 0.04 | 0.01 | 0.04 |

*Note: The first row corresponds to the default configuration of the system.*

Table 5 shows that contrastive learning is essential for preserving semantic structure in the embedding space. Removing contrastive training nearly collapses semantic retrieval performance, yielding MRI values close to zero for both generic and specific evaluations.

Training on the CommonNet corpus consistently improves MRI scores compared to training on CommonGen alone, with especially large gains for generic words. This indicates that WordNet-based supervision helps ground embeddings in abstract semantic concepts rather than in dataset-specific lexical patterns.

The attention mechanism has a comparatively minor effect on semantic retrieval, as configurations with and without attention achieve similar MRI values. This suggests that attention primarily benefits downstream classification, while semantic grounding is largely driven by contrastive learning and semantic supervision.

Overall, these results show that the same design choices that improve forbidden-word detection also help maintain meaningful semantic neighbourhoods in the learned representations. In particular, contrastive

learning combined with explicit semantic supervision enables FoCusNet to act as an effective constraint-filtering module while preserving semantic coherence.

## A.2 LLM prompts

We here provide the prompts we used for the LLM inference:

**Simple Prompt**.

```
Check if the following sentence contains one of the following set of words.
Only answer True or False. Enclose your final answer into '<answer></answer>'.

For instance, if the sentence contains one of the words, answer:
- <answer>True</answer>;
- <answer>False</answer> otherwise.
```

**Chain of Thought**.

```
Check if the sentence contains one of the following set of words.
Only answer True or False. Please, make sure to think step by step.
Enclose your final answer into <answer></answer>.

For instance, if the sentence contains one of the words, answer:
- <answer>True</answer>;
- <answer>False</answer> otherwise.
```

**Best of 3**.

```
For the judges: Check if the following sentence contains one of the following set
of words. Do not include your reasoning process in the answer; Provide a short
explanation (at most 100 words) to justify your answer. Conclude your sentence
with <answer>your answer</answer>, where your answer is either True or False.
```

```
For the final verdict: Give me your final opinion over the verdicts of a jury of
args.n_rounds LLMs. When prompted the following message: <message>, a jury of LLMs
answered: <answer>. What is your final verdict? Enclose your final answer into
<answer></answer>.

For instance, if the sentence contains one of the words, answer:
- <answer>True</answer>;
- <answer>False</answer> otherwise.
```

**LLM as a judge**.

```
Interpret 'expected output' as the ground truth (GT).
The actual output may replace forbidden words. Different wording is allowed.
Goal: FUNCTIONAL SEMANTIC EQUIVALENCE.
The generated sentence should preserve the same overall meaning, scene, intent,
and key facts.
Allow near-synonyms and category-level substitutions when they do NOT change what a
reader would reasonably infer or do.
Pass conditions (must satisfy all):
1) No contradiction: the generated sentence must not contradict GT.
2) No major additions/omissions: it must not add or remove important facts.
Minor stylistic details are okay.
```

```
3) Replacements must preserve ROLE/FUNCTION in context. If the replaced word is an
object/tool, it should be suitable for the same action.
Be permissive for everyday near-synonyms (e.g., bathroom/restroom; modify/change;
shaving cream/shaving lotion) when they preserve the intended action and scene.
Be strict for substitutions that change:
- Names of specific event/holiday/occasion (e.g., 'Remembrance Day' changed to
'Tribute day').
- Safety-critical or instruction-critical details (numbers, dosages, timing,
negation, must/should, etc.).
- Tool appropriateness or specificity in a way that changes the action (e.g.,
scrub brush -> broom for cleaning a toilet).

Scoring guidance:
1.0 = clearly equivalent (perfect paraphrase or safe near-synonym substitutions).
0.8-0.95 = mostly equivalent with slight loss of specificity but still same
intent/scene.
0.5-0.8 = borderline; meaning plausibly shifts or becomes too vague.
<0.5 = meaning changed materially.
```

### A.3    Examples of LLMs answers

Here we report some of the erroneous answers the LLM might return.

**Hallucinations**. Here, the model thinks that the word "dog" is present in the list of forbidden words when processing the sentence "dressed my dog up as a pirate". However, the word is not present – even if the conviction of the model would suggest so.

```
Hallucination: Okay, so I need to figure out which words from the given list are in
the sentence: "dressed my dog up as a pirate." The task is to report all the words
that are contained in the sentence. If none are found, I should just list an empty
list.

First, I'll read the sentence carefully: "dressed my dog up as a pirate." I need to
identify all the individual words in this sentence and see which ones match any from
the provided list.

Let me break down the sentence word by word. The first word is "dressed." I should
check if "dressed" is in the list of words. Looking through the list, I don't see
"dressed" there. So that's one word not present.
Next is "my." Checking the list, "my" isn't there either. Moving on.
The third word is "dog." I'll look for "dog" in the list. Hmm, I don't see it either.
So "dog" isn't present.
Then we have "up." Scanning the list, "up" isn't listed. So that's another word not
found.
The next word is "as." Checking the list, "as" isn't there. So that's not present.

# HERE THE MODEL HALLUCINATE THE EXISTANCE OF THE WORD "PIRATE"
Finally, "a pirate." So "a" and "pirate." Checking the list, "a" isn't there, but
"pirate" is in the list. Wait, let me double-check. The list includes "pirate" as one
of the words. So "pirate" is present.
Wait, but the sentence says "as a pirate," so "a" is part of that phrase, but "a"
isn't in the list. So only "pirate" is in the list.

Let me go through the list again to make sure I didn't miss any. The words provided
```

```
are quite extensive, but I only found "pirate" in the sentence.

So, the only word from the list that's in the sentence is "pirate."
</think>

<answer> True </answer>
```

**Ovethinking #1**. Here, the LLM, prompted to "Think Step by Step", initially gets the correct answer –
i.e., none of the words is present in the sentence. However, it is unsure about the answer, and in a second
pass it eventually self-convince that all words are present.

```
Overthinking: Okay, so I've got this task where I need to check which words from a
given list are present in a specific sentence. The sentence is: "winners podium with
numerals on the sky". The words provided are quite a long list, so I need to make sure
I go through each one carefully.

First, I'll read the sentence again to understand the context. It mentions "winners",
"podium", "numerals", and "sky". I need to see which of the given words match any of
these terms or are related in meaning.

Looking at the list of words, I'll go through them one by one. Starting with
"headlamp" - doesn't seem relevant. "Success" - maybe, but the sentence doesn't
explicitly mention success. "Wrestle" - no, that's unrelated. "Crease" - not here.
"Poultry" - definitely not. "Softball" - nope. "Rodeo" - no. "Bait" - not present.
"Therapy" - doesn't fit. "Whistle" - no. "Navigate" - not in the sentence.
"Aim" - maybe, but not directly. "Attach" - no. "Melody" - no. "Mock" - no. "Pasture"
- no. "Pass" - possibly, but not directly. "College" - no. "Header" - no. "Walker"
- no. "Resident" - no. "Calm" - maybe, but not in context. "Tiger" - no. "Hunter"
- no. "Ambush" - no. "Motorboat" - no. "Position" - maybe, but not clearly.
"Squeeze" - no. "Lease" - no. "Bubble" - no. "Wrap" - no. "Bounce" - no. "Teller"
- no. "Pattern" - no. "Headquarters" - no. "Brightness" - no. "Ballroom" - no.
"Birth" - no. "Statue" - no. "Submarine" - no. "Kiss" - no. "Indoor" - no. "Entry"-no.
"Exhibition" - no. "Proceed" - maybe, but not directly. "Prey" - no. "Puffin" - no.
"Friend" - no. "Swoop" - no. "Asteroid" - no. "Flag" - no.
"Hamburger" - no. "Stool" - no. "Beg" - no. "Resource" - no. "Electricity" - no.
"Backseat" - no. "Anxiety" - no. "Shadow" - no. "Lego" - no.
"Uniform" - no. "Earn" - no. "Shepherd" - no. "Marathon" - no. "Liquid" - no.
"Theme" - no. "Bomber" - no. "Thanksgive" - no. "Request" - no. "Resistance" - no.
"Climb" - no. "Cantilever" - no. "Bag" - no. "Push" - no. "Projector" - no.
"Undergo" - no. "Accompany" - no. "Characteristic" - no. "Nursery" - no. "Salt" - no.
"Mull" - no. "Workman" - no. "Sunset" - no. "Engrave" - no. "Slice" - no. "Crop" - no.
"Clarinet" - no. "Kill" - no. "Blouse" - no. "Purple" - no. "Disappoint" - no.
"Segment" - no. "Shoal" - no. "Memo" - no. "Cite" - no. "Hope" - no. "Deforestation"
- no. "Dial" - no. "Tree" - no. "Militant" - no.

# BEGINNING OF SECOND PASS - WHERE OVERTHINKING HAPPENS
Wait, I think I might have missed "podium". Let me check again. Yes, "podium" is in
the sentence. So that's one word. Also, "winners" is another. "Numerals" is there too.
"Sky" is mentioned as well. Let me make sure none of the other words are in the
sentence. "With" is a preposition, but it's not listed as a word to check. So the
words in the sentence that match the list are winners, podium, numerals, and sky.

Wait, but the original list includes "sky" as one of the words. So I should include
that. So the words are winners, podium, numerals, and sky.
```

```
</think>
<answer> True </answer>
```

**Overthinking #2**. This is an example from the Best of 3 strategy. Two support models agree that the sentence does not contain any words. However, the third model 'advocates for the devil', saying that "adorn" is actually a synonym for "dress", and therefore the sentence shall be labelled invalid. The final judge model trust this third model – as it can provide a convincing explanation – and (erroneously) triggers the alert.

```
Overthinking #2: Give me your final opinion over the verdicts of a jury of 3 LLMs.
When prompted the following message: Task: Check if the following sentence contains
one of the following set of words. Do not include your reasoning process in the anser;
Provide a short explanation (at most 100 words) to justify your answer.
Conclude your sentence with <answer>your answer</answer>, where your answer is either
True or False.
Sentence: dressed my dog up as a pirate .
Words: <LIST_OF_WORDS>

# REPORTING ANSWERS FROM JUDGES
A jury of LLMs answered:

Judge 0: <think> Okay, so I need to check if the sentence "dressed my dog up as
a pirate" contains any of the given words. Let me go through each word one by one to
make sure I don't miss any.
First, the sentence is about dressing a dog as a pirate. So, the main elements are
dressing, dog, pirate. Now, looking at the list of words, I see words like "adorn,"
which means to decorate or dress up. That's a possibility because the sentence is
about dressing the dog.

Next, I'll check the other words to be thorough. "Container" doesn't fit here.
"Length" and "hairstyle" are unrelated. "Mountain," "lesson," "shop," "boutique,"
"glider," "iron," "steal," "orangutan," "overall," "blur," "lava," "equestrian,"
"exert," "hear," "scuba," "slice," "manufacturing," "personality," "decor,"
"cemetery," "plantation," "ear," "excite," "argue," "inscription," "cook," "deity,"
"pallet," "skyline," "hose," "chromosome," "awning," "contrail," "brand," "right,"
"medina," "catwalk," "sheer," "berth," "lamb," "complain," "impress," "ignore,"
"onion,"  "thrive," "smell," "guitarist," "effect," "header," "aim," "ask," "dome,"
"panini," "headdress," "table," "adorn," "evacuate," "sparkle," "girlfriend,"
"synchrony," "export," "tag," "puff," "school," "fiddle," "queue," "beam," "medal,"
"intensify," "distance," "alm," "bullfight," "midair," "mangrove," "bless," "wall,"
"mow," "universe," "candle," "galaxy," "reel," "soar," "worship," "sneeze," "gallop,"
"clank," "chimney," "helmet," "allow," "leather," "embrace," "iceberg," "lantern,"
"stride," "vacation," "wallow."

Wait, I already found "adorn" as a possible match. Let me make sure there's no other
word that might fit. "Adorn" is the only one that relates to dressing or decorating
something. The sentence is about dressing the dog, so "adorn" fits perfectly. The
other words don't seem to connect at all. So, the sentence does contain one of the
specified words.
</think>
The sentence contains the word "adorn," which means to decorate or dress something.
Since the sentence is about dressing a dog as a pirate, "adorn" fits the context.
<answer>True</answer>

Judge 1: <think> Alright, so I need to check if the sentence "dressed my dog up as a
```

pirate" contains any of the given words. Let me go through each word one by one. First,
I'll read the sentence carefully. It says the user dressed their dog up like a pirate.
So, the main actions here are dressing and being a pirate.

Looking at the list of words, I see words like 'dress' or 'clothing' aren't present.
The sentence mentions 'dog' and 'pirate', but those aren't in the list either. I'll
go through each word from the list to make sure I don't miss anything. Words like
'massage', 'container', 'length', 'hairstyle' - none of these are in the sentence.
'Mountain', 'lesson', 'shop', 'boutique' - nope. 'Glider', 'iron', 'steal',
'orangutan' - still nothing. 'Overall', 'blur', 'lava', 'equestrian', 'exert',
'hear', 'scuba', 'slice', 'manufacturing', 'personality', 'decor', 'cemetery',
'plantation', 'ear', 'excite', 'argue', 'inscription', 'cook', 'deity', 'pallet',
'skyline', 'hose', 'chromosome', 'awning', 'contrail', 'brand', 'right', 'medina',
'catwalk', 'sheer', 'berth', 'lamb', 'complain', 'impress', 'ignore', 'onion',
'thrive', 'smell', 'guitarist', 'effect', 'header', 'aim', 'ask', 'dome', 'panini',
'headdress', 'table', 'adorn', 'evacuate', 'sparkle', 'girlfriend', 'synchrony',
'export', 'tag', 'puff', 'school', 'fiddle', 'queue', 'beam', 'medal', 'intensify',
'distance', 'alm', 'bullfight', 'midair', 'mangrove', 'bless', 'wall', 'mow',
'universe', 'candle', 'galaxy', 'reel', 'soar', 'worship', 'sneeze', 'gallop',
'clank', 'chimney', 'helmet', 'allow', 'leather', 'embrace', 'iceberg', 'lantern',
'stride', 'vacation', 'wallow'.

After going through each word, I don't see any that match the sentence.
The sentence uses 'dressed' and 'dog', but those aren't in the list. The closest might
be 'adorn', but that's not present either. So, the sentence doesn't contain any of
the specified words.
</think>
<answer>False</answer>

**Judge 2:** <think>
Okay, so I need to check if the sentence "dressed my dog up as a pirate" contains any
of the given words. Let me list out the words first to make sure I don't miss any.
The words are: massage, container, length, hairstyle, mountain, lesson, shop, boutique,
glider, iron, steal, orangutan, overall, blur, lava, equestrian, exert, hear, scuba,
slice, manufacturing, personality, decor, cemetery, plantation, ear, excite, argue,
 inscription, cook, deity, pallet, skyline, hose, chromosome, awning, contrail, brand,
 right, medina, catwalk, sheer, berth, lamb, complain, impress, ignore, onion, thrive,
 smell, guitarist, effect, header, aim, ask, dome, panini, headdress, table, adorn,
 evacuate, sparkle, girlfriend, synchrony, export, tag, puff, school, fiddle, queue,
 beam, medal, intensify, distance, alm, bullfight, midair, mangrove, bless, wall,
 mow, universe, candle, galaxy, reel, soar, worship, sneeze, gallop, clank, chimney,
 helmet, allow, leather, embrace, iceberg, lantern, stride, vacation, wallow.

Hmm, the sentence is about dressing a dog as a pirate. So, the main elements are
"dog," "dress," "pirate." Now, looking through the list, I don't see "dog" listed,
so that's a point against it. But wait, the words are about checking if any of them
are present in the sentence, not necessarily the other way around.

Wait, the sentence is "dressed my dog up as a pirate." So the words used are
"dressed," "my," "dog," "up," "as," "a," "pirate." None of those words are in the
provided list. Let me double-check the list again. The words are all nouns,
I think. So "dressed" is a verb, "dog" is a noun, but "dog" isn't in the list. So
the sentence doesn't contain any of the specified words. Therefore, the answer
should be False.

```
</think>
<answer>False</answer>

# BACK TO THE FINAL VERDICT

What is your final verdict? Just answer True or False: Ensure to exclude your final
answer into <answer></answer>. For instance, if the sentence contains one of the
words, answer <answer>True</answer>; <answer>False</answer> otherwise.

<think>\nAlright, so I need to determine if the sentence "dressed my dog up as a
pirate" contains any of the specified words. Let me go through each word in the
list to see if any match.\n\nFirst, the sentence is about dressing a dog as a pirate.
The key elements are "dress," "dog," and "pirate." Now, looking at the list of words
provided: massage, container, length, hairstyle, mountain, lesson, shop, boutique,
glider, iron, steal, orangutan, overall, blur, lava, equestrian, exert, hear, scuba,
 slice, manufacturing, personality, decor, cemetery, plantation, ear, excite, argue,
 inscription, cook, deity, pallet, skyline, hose, chromosome, awning, contrail,
brand, right, medina, catwalk, sheer, berth, lamb, complain, impress, ignore,
onion, thrive, smell, guitarist, effect, header, aim, ask, dome, panini, headdress,
table, adorn, evacuate, sparkle, girlfriend, synchrony, export, tag, puff, school,
fiddle, queue, beam, medal, intensify, distance, alm, bullfight, midair, mangrove,
bless, wall, mow, universe, candle, galaxy, reel, soar, worship, sneeze, gallop,
clank, chimney, helmet, allow, leather, embrace, iceberg, lantern, stride,
vacation, wallow.\n\nAfter reviewing each word, I notice that "adorn" is present
in the list. The word "adorn" means to decorate or dress something. Since the sentence
is about dressing a dog, "adorn" fits the context. Therefore, the sentence does
contain one of the specified words.\n<answer>True</answer>'
```

