# OpenReview forum: "Large-Scale Constraint Generation - Can LLMs Parse Hundreds of Constraints?"
_TMLR — Rejected by TMLR_

### Review · Reviewer_8dsW · 2025-11-11

**Summary Of Contributions:**

This paper explores the constrained generation capabilities of large language models. In particular, this paper introduces LSCG---Large-Scale Constraint Generation, relating Test Steering Strategies techniques to LSCG.

Three test steering strategies are reviewed, including: Simple prompt, chain of thought, and best of N judges. The paper trains FoCusNet to perform a binary classification task over individual constraints.

The methodology involves engineering Words Checker, where an LLM must classify a sentence as valid or invalid based on a dynamically provided list of forbidden words. The Words Checker is explicitly designed to study the impact of an increasing number of forbidden words on LLM performance.

The experiments involve the results of the above three test steering strategies on the Words Checker. The experiments are run on several open-source models, including Llama and DeepSeek models. The main findings include:
- For all models, as the number of forbidden words increases, accuracy drops by around 30%.
- Chain of thought does not improve over basic prompting when the number of forbidden words is small.

Pros:
- The topic is interesting as it proposes to better understand the mechanism of instruction-following tasks through controlled benchmarks.

Cons:
- The paper is poorly written and hard to follow. There are many typos throughout the paper, making it difficult to read.
- I find that this paper is rather preliminary; although the ideas are interesting, I did not find enough results or experiments to convince me regarding the proposed LSCG framework.

**Audience:**

No

**Audience Explanation:**

I think that this paper is interesting, but the way it's written is very confusing.

**Claims And Evidence:**

No

**Claims Explanation:**

My understanding about this paper is that it constructs the Words Checker task from previous CommonGen benchmark. Then, it trains a customized model (FoCusNet) on this dataset (using a combination of pre-training and contrastive learning techniques). Finally, it compares test steering techniques to FoCusNet results.

The contributions, however, are not very clear to me based on this:
- LSCG has a "novel" problem formulation itself does not appear to be a contribution ---the Words Checker problem makes more sense. However, this dataset is constructed from another existing dataset (Lin et al. (2020)).
    - If I take this as a dataset contribution, then it wasn't clear how much of the data was created by the authors or how much of it was from the earlier benchmark/paper. It also wan't clear how large was the dataset, how much information does it contain. Whether the dataset construction procedure might lead to different kinds of conclusions in downstream studies.
- The relatively small, dedicated model on the Words Checker dataset is fine. But here the authors did not run any comparative studies on this model. What about standard architectures like CNN, transformers, etc.? How do these architectures compare to FoCusNet?
- Finally, the experimental comparisons are somewhat vague, and it wasn't clear what sort of new insights these studies are providing to the community.

**Requested Changes:**

- Overall, I find that the paper is very hard to read, and it wasn't clear what the key claims made in the paper are. The organization needs improvement.

- The references are misformatted, and they are missing a "(", ")" as part of the citation.

- There are many typos throughout the paper. E.g., in Table 1, "Chain of Though" -> "Chain of Thought"

- In the beginning of Section 5, here it would be useful to have an overview of the experiments section.

- The list of contributions needs to be better stated. As it is, I believe the paper needs an extensive revision and a significantly expanded methodology/experiments section.

---

> ### Author Response · Authors · 2026-01-19
> **Addressing reviewer's comments**
>
> We thank the reviewer for the detailed feedback. We agree that the previous version of the paper had clarity and organization issues, and we have carried out a careful rewrite to address them. We corrected typographical and citation-formatting errors, clarified terminology, and improved the overall structure. We also added a brief overview at the beginning of the experimental section to explain the purpose of each group of experiments, and we rewrote the contribution list to more clearly state the paper’s key claims.
>
> We clarified the scope and positioning of the work. In the revised version, Words Checker is explicitly framed as a constraint-identification benchmark, and LSCG is presented as a problem setting rather than as a standalone dataset contribution. We now clearly describe how the dataset is derived from CommonGen, specifying what is reused, what is newly constructed, and providing details on dataset size and construction.
>
> To strengthen the empirical evaluation, we expanded the experiments in several ways. We added a non-LLM baseline based on classical string matching with stemming and lemmatization, which achieves perfect performance and helps isolate the failure mode of LLMs. We also included an ablation study of FoCusNet in the appendix, analyzing the effect of removing contrastive training, attention, semantic supervision, and changing the classification head.
>
> Finally, to address concerns about the lack of generation-based evaluation, we introduced a second task, Language Moderator, which requires constrained generation while preserving semantics. The results show that the same constraint-identification bottleneck persists - and becomes more severe - when generation is required, supporting the paper’s central argument.
>
> We believe these revisions substantially improve clarity, positioning, and experimental support, and make the paper’s contributions more accessible to the TMLR audience.

---

### Review · Reviewer_P8yN · 2025-11-29

**Summary Of Contributions:**

The paper introduces Large-Scale Constraint Generation (LSCG), a setting where an LLM must obey a large number (e.g. more than 100) of fine-grained constraints when solving a task. Instead of a few explicit instructions, the model receives a long list of generic constraints and must implicitly identify which ones matter and follow them.
For such problem, the authors design a concrete instance called Words Checker, in which the task is to classify whether the sentence as valid or invalid based on the given sentence and constraints.
They evaluate several test-time steering strategies on multiple LLMs with approaches of simple prompt, chain-of-thought, best-of-N, and the proposed FoCusNet. On DeepSeek-R1, FoCusNet improves accuracy over simple prompt/CoT/best-of-3 for 500-1000 forbidden words and maintains higher precision, especially in high-constraint regimes.

**Audience:**

Yes

**Audience Explanation:**

The paper focuses on a very realistic LLM related problem: modern LLM applications often must respect long lists of constraints, such as policies, style guides, safety guidelines, documentation.

**Claims And Evidence:**

No

**Claims Explanation:**

Although the paper is well written and organized, I think it is not convincing whether the claim and results are generalizable. First of all, I think tha paper is limited to a simple empirical validation for the claim, while the claim is very broad and general (e.g., travel guides, documentation). The main task (word checker) is binary classification task, which is even not generation task. It makes the framing “Large-Scale Constraint Generation” feel somewhat overstated. As a result, it is unclear how well the conclusions and FoCusNet would transfer to more realistic, messy constraint sets.
Secondly, the evaluation of FoCusNet is relatively narrow. The main detailed FoCusNet results (Table 2) are all on one model. It would strengthen the paper to show at least some FoCusNet results with different LLM families or sizes. Furthermore, ablation of FoCusNet is missing. For example, the paper should include results without contrastive learning, attention, or with different classifier (not the random forest) to see which part is the most effective for the problem.

**Requested Changes:**

Please find the above comments. Especially, the paper needs more realistic problems (including generation tasks) with broader LLM models, and ablation of FoCusNet. Given those results and analysis, I believe the paper would be more stronger.

---

> ### Author Response · Authors · 2026-01-19
> **Addressing reviewer's comments**
>
> We sincerely thank the reviewer for their thoughtful and constructive feedback.
>
> Following the reviewer’s suggestions, we substantially expanded the experimental section to better assess the generalizability of our claims. First, we extended the comparison between FoCusNet and traditional test-steering strategies to an additional model, DeepSeek-70B. This allows us to evaluate whether the observed trends persist across LLMs of different scales and architectures. The results confirm that the improvements introduced by FoCusNet consistently generalize beyond a single model configuration.
>
> Second, we added a comprehensive ablation study in the appendix to analyze the contribution of FoCusNet’s individual components. In particular, we investigate the impact of contrastive learning, the attention mechanism, the use of a semantic training corpus, and alternative classifier choices. These experiments clarify which design decisions are most influential for the overall performance and robustness of the method.
>
> Finally, to address concerns regarding task diversity and the absence of generation-based evaluation, we introduce a second task, Language Moderator. Unlike the original Word Checker, which is a binary classification task, Language Moderator is a constrained generation task. Given an input sentence containing forbidden words, the model must generate a new sentence that (i) removes the forbidden content and (ii) remains semantically equivalent to the original sentence. This task therefore introduces an additional layer of logical complexity beyond constraint identification, namely controlled paraphrase generation.
>
> Importantly, the results show that even this modest increase in task complexity leads to severe performance degradation when constraint identification is not explicitly addressed. Using a simple prompting strategy, models are able to correct only 3.9% of invalid sentences with an 8B model and 17.8% with a 70B model. These failures occur despite the fact that the generation step itself is relatively straightforward, highlighting that the primary bottleneck lies in identifying and selecting the relevant constraints rather than in performing the generation.
>
> From this perspective, the Word Checker task can be viewed as an upper-bound diagnostic setting that isolates the constraint identification problem and allows us to measure it systematically. The Language Moderator task then demonstrates how this bottleneck propagates - and is amplified - once additional reasoning or generation is required, as would be the case in realistic applications. Notably, incorporating FoCusNet significantly mitigates this issue, improving performance from 3.9% to 11.9% for the 8B model and from 17.8% to 34% for the 70B model. Still, results are far from the performance one would expect in real world scenarios.
>
> Taken together, these two case studies are sufficient to capture how LLMs behave in practical, constraint-heavy scenarios such as documentation editing, moderation, or instruction-following. If the constraint parsing and identification problem is not addressed at this foundational level, any downstream task that requires additional logical or generative reasoning will inevitably inherit - and exacerbate - these failures.

---

### Review · Reviewer_14Sz · 2025-12-23

**Summary Of Contributions:**

This paper studies the capability of LLMs to parse and follow a large number of constraints, and investigates how performance degrades as the number of constraints increases. To this end, the authors introduce a synthetic benchmark, termed Words Checker, in which a model must label a sentence as valid or invalid depending on whether it contains any forbidden words from a provided list, whose size varies across 10, 100, 500, and 1000. Three common model-steering techniques are evaluated and compared against the proposed approach, FoCusNet, which is an auxiliary module that operates in conjunction with an LLM to reduce the full constraint set to a smaller subset, with the aim of improving robustness. Overall, the paper is well written and easy to follow. The motivation is clear, as real-world tasks often involve lengthy guidelines and numerous constraints. However, the proposed approach, evaluation setup, and resulting contributions are highly limited (see comments below), and there is substantial room for improvement.

**Additional Comments:**

Minor comments:

1) Abstract: "number constraints" -> "number of constraints"
2) Section 4: "Ratio behind Word Checker"  -> "Rationale behind Word Checker"
3) Table 1: "Chain of Though" -> "Chain of Thought"

**Audience:**

Yes

**Audience Explanation:**

The community working on instruction-following of LLMs.

**Claims And Evidence:**

No

**Claims Explanation:**

Below are my concerns and comments.

1) The paper's main empirical finding demonstrates that accuracy and precision of LLMs degrades as forbidden-word list grows. There is already a large body of literature showing that LLMs struggle with multi-constraint instruction following and proposing ways to improve it (decomposition, critique–refine loops, etc.). There are also various related benchmarks such as FollowBench (https://arxiv.org/abs/2310.20410), ComplexBench (https://arxiv.org/abs/2407.03978), and CFBench (https://arxiv.org/abs/2408.01122). None of these are discussed in the related works or included in the evaluation.

2) The proposed benchmark instance, Words Checker is essentially classification, not constrained generation as framed in the paper. This gap needs an explicit justification or rewriting.

3) Claims like "LSCG examines whether LLMs can replicate humans’ practical intelligence" are overly strong relative to the current content of the paper.

4) The introduction states that the accuracy of a distilled 8 B model drops down to 27.8 % under test strategies, yet the numbers in Table 2 convey a different picture. This needs reconciliation.

**Requested Changes:**

Below are my recommendations on how to address the aforementioned concerns.

1) Either reposition Words Checker as a constraint-parsing benchmark (not "generation") or add a true generation task instance where outputs must satisfy a large constraint list.

2) Update the literature review section to include the aforementioned benchmarks and clearly articulate the added contributions of this work.

3) Add non-LLM (e.g., classical string matching) and retrieval baselines to strengthen the evaluation and the interpretation of the results.

4) Broaden LSCG beyond word lists to justify the "general constraints" claim (e.g., policy adherence task with bulleted constraints or something along those lines)

5) Provide ablation studies for FoCusNet (e.g., without RF, without contrastive training)

---

> ### Author Response · Authors · 2026-01-19
> **Addressing reviewer's comments**
>
> We thank the reviewer for the careful reading of the paper and for the constructive suggestions. Below we address each concern and describe the corresponding changes made in the revised manuscript.
>
> - Related work and positioning: We expanded the literature review to include FollowBench (Jiang et al., 2024), ComplexBench (Wen et al., 2024), and other recent benchmarks on multi-constraint instruction following. We now clearly position our contribution relative to these works. In particular, while FollowBench and ComplexBench study degradation as a small number of explicitly relevant constraints (up to five or six) are composed, our work focuses on a complementary and underexplored regime: settings in which the constraint pool is orders of magnitude larger (hundreds to thousands), while only a small subset (at most four) is contextually applicable and capable of invalidating an otherwise correct response. This distinction is now explicitly stated and motivates our focus on constraint identification rather than constraint composition.
> - Words Checker as classification vs. generation: We agree that Words Checker is fundamentally a classification task. In the revised version, we explicitly reposition it as a constraint-parsing benchmark rather than a constrained generation task, and we clarify its role as a diagnostic proxy. We also added a discussion in the related-work section highlighting prior work that similarly uses classification tasks to probe generation-relevant capabilities (e.g., Measuring Massive Multitask Language Understanding), thereby justifying this methodological choice.
> - Addition of a true generation task: To address the gap between classification and generation, we introduce a new task, Language Moderator. In this task, the model must generate a revised sentence that removes forbidden words while preserving the original semantics. This task requires both constraint identification and controlled generation, directly aligning with the constrained generation framing. The results show that even this modest increase in task complexity leads to severe performance degradation when constraint identification is not explicitly addressed, reinforcing the central thesis of the paper.
> - Baselines beyond LLMs: We added a non-LLM baseline based on classical string-matching techniques augmented with stemming and lemmatization. As expected, this baseline achieves perfect performance on both Words Checker and Language Moderator. This serves two purposes: (i) it validates the correctness and evaluability of the benchmarks, and (ii) it highlights that the difficulty faced by LLMs is not inherent to the task itself, but arises from failures in constraint parsing and identification.
> - Ablation study of FoCusNet: We added a detailed ablation study in the appendix, evaluating FoCusNet without contrastive training, without attention, and with alternative classification heads in place of the random forest. These results clarify the contribution of each design component and directly address the reviewer’s concern.
> - On broadening LSCG beyond word lists: We acknowledge the reviewer’s suggestion to include an additional LSCG task with more complex, structured constraints (e.g., policy adherence). However, we deliberately limit the scope of this work to tasks with unambiguous, automatically verifiable evaluation. Our results already demonstrate that LLM performance degrades sharply even in extremely simple settings with clear-cut correctness criteria. This suggests that constraint identification is a fundamental bottleneck that is likely to persist - and potentially worsen - in more complex and realistic scenarios. Designing and validating additional tasks with reliable evaluation protocols would require careful methodological treatment and is therefore left as future work.

---

### Author Response · Authors · 2026-01-19
**Revised Paper addressing reviewers' comments**

Dear Reviewers,

Thank you for the valuable feedback. We have uploaded a revised version of the paper that addresses all the points raised in the previous reviews. In particular, the new version:

- Has been substantially revised and carefully edited to improve clarity and presentation. We better position LSCG within the existing literature by incorporating the suggested references and by clarifying why the problem is both important and currently under-explored.
- Introduces an additional LSCG task, termed Language Moderator. In this task, the LLM is given a sentence and a list of forbidden words and is required to generate a revised sentence that removes any forbidden words while preserving the original meaning. We show that, under large-scale constrained generation, this additional semantic-preservation requirement leads to a significant performance degradation, which is partially mitigated by FocusNet.
- Expands the experimental evaluation by including additional models, notably an analysis of DeepSeek 70B in Table 2. We also augment Table 2 with an additional steering strategy (Elicit Reasoning) and a RAG baseline. The results show that FocusNet consistently achieves the best performance, with similar trends observed across both smaller and larger models.
- Provides a thorough ablation study of FocusNet, highlighting its key components. In particular, we find that contrastive loss training is the primary factor driving performance gains. The use of attention mechanisms and a semantic corpus provides moderate additional benefits, while the choice of classification head (Random Forest or KNN) yields comparable performance.
- Includes an anonymized version of the code and datasets to support reproducibility.

---

### Decision · Action_Editor_67Ay · 2026-04-08

**Recommendation:** Reject

**Additional Comments:**

The work can be improved by providing:
- other genuine generation tasks under constraints;
- more results by major LLMs, especially the ones with improved reasoning capabilities: e.g., GPT-OSS-120B, GLM-4.7, Kimi K2 Thinking, MiniMax-M2.1, Qwen3-Next-80B-A3B.

It would also be interesting to report results obtained with popular online  LLMs, such as chatGPT (based on GPT-5.3), Gemini 3, Claude Opus 4.1.

**Audience:**

No

**Audience Explanation:**

The covered topic is of potential interest to TMLR's audience interested in applications involving LLMs. Knowing the limitations of LLMs is helpful to avoid spending time and resources to achieve results that are not reachable.

**Claims And Evidence:**

No

**Claims Explanation:**

Although the authors have provided an improved version of the work, especially by adding a new dataset covering in a better way the generation task, considering the 70B version of DeepSeek, and providing an ablation study of the FoCusNet, the claims of the paper are still too strong with respect to the broad scope of the investigated questions. Two our or three reviewers are still not convinced that the contribution of the paper is enough to justify publication, although the covered topic seems to be interesting.

**Resubmission Of Major Revision:**

The authors may consider submitting a major revision at a later time.